# Prevalence and predictors of obesity-related cancers among racial/ethnic groups with metabolic syndrome

Shanada Monestime[1]*, Bettina Beech[2], Dulcie Kermah[3], Keith Norris[4]

**1** Department of Pharmacotherapy, University of North Texas System College of Pharmacy, Fort Worth, Texas, United States of America, **2** University of Houston, College of Medicine, Houston, Texas, United States of America, **3** Charles R. Drew University of Medicine and Science, Los Angeles, California, United States of America, **4** UCLA Division of General Internal Medicine and Health Services Research, Los Angeles, California, United States of America

* Shanada.monestime@gmail.com

## Abstract

**Data Availability Statement:** Data availability statement: The data that support the findings of this study are openly available in NHANES at https://wwwn.cdc.gov/nchs/nhanes/, reference number 1999-2014.

### Background

Obesity-related cancer (ORC) is associated with higher amounts of body fat, which could increase the risk of developing cardiovascular disease (CVD). A significant factor associated with CVD is metabolic syndrome (MetS), and MetS prevalence differs by race/ethnicity. The purpose of this study was to compare the prevalence and predictors of ORCs by race/ethnicity among adults (>18) with MetS.

### Methods

This was a retrospective, cross-sectional study using data from the 1999–2014 National Health and Nutrition Examination Survey (NHANES). A chi-square test was performed to determine differences in ORC prevalence between non-Hispanic White (NHW), non-Hispanic Black (NHB), and Hispanic participants with MetS. A multivariate logistic regression was used to evaluate predictors (race, sex, income, insurance, education, marital status, and smoking status) of ORC among adults with MetS.

### Results

Of the 1,554 adults, the prevalence of ORC was 30.6% among NHWs, 51.3% in NHBs, and 54.1% in Hispanics ($p$ = <0.001). Females were 6.27 times more likely to have an ORC compared to males (95% CI = 4.95–14.11). Compared to NHWs, NHBs were 2.1 times more likely to have an ORC (95% CI = 1.40–3.38); and Hispanics were 2.5 times more likely (95% CI = 1.39–4.77). For every 1-year unit increase in age, the odds of ORC increased by 3% (95% CI = 1.00–1.05).

**Funding:** SM, BB, DK, NK-This research was supported by a grant from the National Heart, Lung, and Blood Institute to the University of Mississippi Medical Center (Grant #2R25HL126145 – MPIs Beech and Norris). The funders had no role in study design, data collection and analysis, decision to publish, or preparation of the manuscript.

**Competing interests:** The authors have declared that no competing interests exist.

## Conclusions

Among NHANES participants with MetS, the prevalence of ORCs was significantly higher in NHBs and Hispanics, females, and older adults with MetS. Future studies, by race/ethnicity, are warranted on mortality risk of persons with MetS and ORC.

## Introduction

Obesity-related cancers (ORCs) account for 40% of all cancer diagnoses in the United States and are associated with higher health expenditures compared to non-ORCs [1,2]. ORCs are hypothesized to have biological properties that are related to poor prognosis [3]. These types of cancers are associated with higher body mass index (BMI $\geq$25 kg/m$^2$) and include meningioma, bladder, esophageal, kidney, endometrial, ovarian, thyroid, liver, gallbladder, stomach, pancreatic, colorectal, and certain blood cancers [1,4–7]. It is important to note that not all adults who have an ORC are overweight or obese. However, among adults with cancer and obesity, an association exists with a reduced likelihood of undergoing cancer screening, difficulty in detecting cancer during a physical exam, and receiving suboptimal doses of chemotherapy [3]. Also, numerous sequelae are associated with having ORCs such as having an increased risk of type 2 diabetes, hypertension, coronary heart disease, or a combination of these. A concern exists that these risk factors can lead to cardiovascular disease (CVD) [8], which remains the leading cause of death in cancer survivors [9].

Risk factors for cardiovascular disease tend to cluster into interrelated groups of conditions commonly known as metabolic syndrome (MetS), which has been associated with an increased risk of CVD in adults with cancer [10]. MetS is defined by having at least three of the following: abdominal obesity, diabetes, hypertension, hypertriglyceridemia, or low levels of high-density lipoprotein (HDL). Lifestyle interventions, including increasing physical activity and reducing excessive caloric intake, are widely accepted as safe and effective treatments for MetS [11]. Furthermore, excessive caloric intake from frequent consumption of soft drinks has been consistently correlated with a higher waist circumference and a modest increase in risk for developing an ORC among adults [12]. In addition, the risk of breast cancer, which is the most common ORC, was lower among the most physically active women compared to the least active women in the majority of studies evaluated in a systematic review [13]. The high prevalence of sedentary lifestyle and associated weight gain seen in the United States may be a contributing factor to the high rates of MetS and cardiovascular morbidity observed in cancer survivors [11]. Therefore, a critical need exists to understand ORC as it relates to a diagnosis of MetS.

Previous studies have identified the prevalence of ORC to be higher in non-Hispanic Blacks and Hispanics compared with non-Hispanic Whites [1,2]. However, prior studies did not control for sociodemographic and lifestyle factors that can differ across race/ethnicity and that may provide further insights on why select groups may be at risk for developing CVD. Therefore, the primary objective of this study was to compare, by race/ethnicity, the prevalence of ORCs among a nationally representative sample of adults with MetS. The secondary objective was to identify predictors associated with adults who have MetS concurrently with an ORC.

## Methods

### Study design

Data were obtained from the 1999–2014 National Health and Nutrition Examination Survey (NHANES), a series of cross-sectional surveys of adults in U.S. households. Home interviews

were conducted to collect self-reported information, including, but not limited to, demographic data, socioeconomic data, and cancer diagnosis, followed by an extensive physical examination and blood collection at a mobile examination center [14,15]. The survey uses a complex, multistage probability design to provide a nationally representative sample of the U. S. civilian noninstitutionalized population and allows for a high level of generalizability to the nation's population. Further details of the NHANES survey design, questionnaires, and examination methods are described elsewhere [16]. The present study was not reviewed by the Institutional Review Board as the data analyzed are de-identified and publicly accessible.

## Study population

Adult NHANES participants were included in the analytic sample if they self-reported as non-Hispanic White, non-Hispanic Black, or Hispanic; were greater than 20 years old; and were diagnosed with comorbid MetS and cancer. Pregnant women were excluded because of increased waist circumference and potential pregnancy-related metabolic changes. Respondents who self-identified as "Mexican American" and "other self-identified Hispanics" were grouped as Hispanics. All other non-Hispanic participants were categorized based on their self-reported race.

Adults categorized as having MetS (at least three of the following: hypertension, diabetes, abdominal obesity, hypertriglyceridemia, or low levels of HDL) were identified using related NHANES questions and laboratory values defined by the National Cholesterol Education Program Adult Treatment Panel III (NCEP ATP III) guidelines. Hypertension was defined by a measurement of 130/85 mmHg or greater and/or a positive response to the question, "Are you now taking prescribed medicine (for blood pressure)?" Diabetes mellitus was defined by a fasting blood sugar over 100 mg/dL and/or a positive response to the question, "Are you now taking diabetic pills to lower blood sugar?" or "Are you taking insulin now?" Abdominal obesity was defined by a waist circumference over 40 inches (men) or 35 inches (women). Hypertriglyceridemia was defined as a fasting triglyceride level over 150 mg/dL, and low HDL cholesterol levels were defined by a fasting HDL cholesterol level less than 40 mg/dL (men) or 50 mg/dL (women).

Among participants with MetS, we identified those with one or more cancers. We determined whether adults had an ORC compared to a non-ORC based on their response to the following question, "Have you ever been told by a doctor or other health professional that you had cancer or a malignancy of any kind?" Presence of an ORC was determined if the respondent had a diagnosis of at least one the following cancers: brain, bladder, esophageal, kidney, endometrial, ovarian, thyroid, breast, liver, gallbladder, stomach, pancreatic, or colorectal [4,5]. All other cancers were defined as non-ORCs (bone, blood, cervical, head and neck, leukemia, liver, lung, lymphoma, melanoma, nervous system, prostate, rectum, skin cancer (non-melanoma), testicular, and other cancers). An affirmative response to having cancer was followed by an opportunity to specify up to three different cancer diagnoses. If one of three cancers was obesity related, the survey participant was categorized as having an ORC, even if the other two cancers were non–obesity related. Respondents who had non-ORCs were defined by not having a prior or current diagnosis of an ORC.

## Study variables

Covariates included sociodemographic factors, insurance status, and smoking status. Sociodemographic factors included five variables. Age was measured as a continuous variable, and sex was coded as a dichotomous variable that indicated whether a participant was female or male. Education status was stratified by whether participants completed less than high school, a high

school diploma/general education diploma (GED) or equivalent, or some college or above. Income was determined by the family's total annual income of either less than $35,000; $35,000 to $74,999; or $75,000 and above. Marital status was stratified by being a widow, divorced, separated, never married, married, or living with a partner. Insurance status was coded as a dichotomous variable, indicating whether a participant was insured. Lastly, smoking status was stratified by the participant either smoking every day, some days, or not at all.

## Statistical analysis

All statistical analyses were conducted using SAS 9.4. Statistical significance was assessed with two-tailed tests and $\alpha = 0.05$. A chi-square test was performed to determine differences in ORC prevalence among non-Hispanic Whites, non-Hispanic Blacks, and Hispanics diagnosed as having both MetS and cancer. A multivariable logistic regression was used to evaluate possible predictors (race, sex, income, insurance, education, marital status, and smoking status) of ORC among adults with MetS. Data from the regression model were reported as the odds ratio (OR), 95% confidence interval (CI), and associated $p$-value for the specific characteristic. We calculated the mean and SD for continuous variables. For categorical variables, we obtained the frequencies.

## Results

### Patient characteristics

Among 1,554 adults diagnosed with MetS, a higher prevalence of non-ORCs ($n = 974$, 67%) existed compared with ORCs ($n = 580$, 33%). Among the 580 adults with ORC, the mean age was 64.2 years (SE ± 0.76), and most were female ($n = 448$, 81.6%), non-Hispanic White (n = 376, 82.9%), married ($n = 286$, 57.4%), and non-smokers ($n = 205$, 71.3%). Nearly half of the study participants had some college education or above ($n = 229$, 48.1%), nearly a third had income less than $35,000 ($n = 177$, 32%), and most had some form of health insurance ($n = 544$, 95.3%). The top five ORCs in this sample were breast, colorectal, uterine, bladder, and ovarian cancer (Fig 1). For adults with non-ORCs, the mean age was 67.5 years (SE ± 0.46), and most of the adults were male ($n = 634$, 60.1%), non-Hispanic White ($n = 759$, 92.38%), married ($n = 623$, 67.2%), non-smokers ($n = 456$, 72.4%), had some college-education or above ($n = 497$, 58.2%), and had health insurance ($n = 917$, 93.4%). Annual income was equally distributed across categories of less than less than $35,000; $35,000 to $74,999; or $75,000 and above. All baseline characteristics differed between obese and non-obese groups with the exception of insurance status ($p = 0.29$) and smoking status ($p = 0.5$; Table 1).

### Weighted prevalence of obesity-related cancer by race/ethnicity

Of the adults with comorbid MetS and cancer, 239 (6.32%) were non-Hispanic Blacks, 189 (4.40%) were Hispanics, and 1126 (89.27%) were non-Hispanic Whites. The prevalence of ORC among non-Hispanic Blacks, Hispanics, and Whites was 51%, 54%, and 31%, respectively ($p = <0.001$). However, for non-ORCs, the prevalence was higher in non-Hispanic Whites (69%; Fig 2).

### Predictors of obesity-related cancers among adults with comorbid metabolic syndrome and cancer

Of the predictors analyzed using a multivariable logistic regression approach, three of the eight variables were significantly associated with ORC in adults with comorbid MetS and cancer. Females were 6.27 times more likely to have an ORC than males (95% CI = 4.95–14.11; $p = $

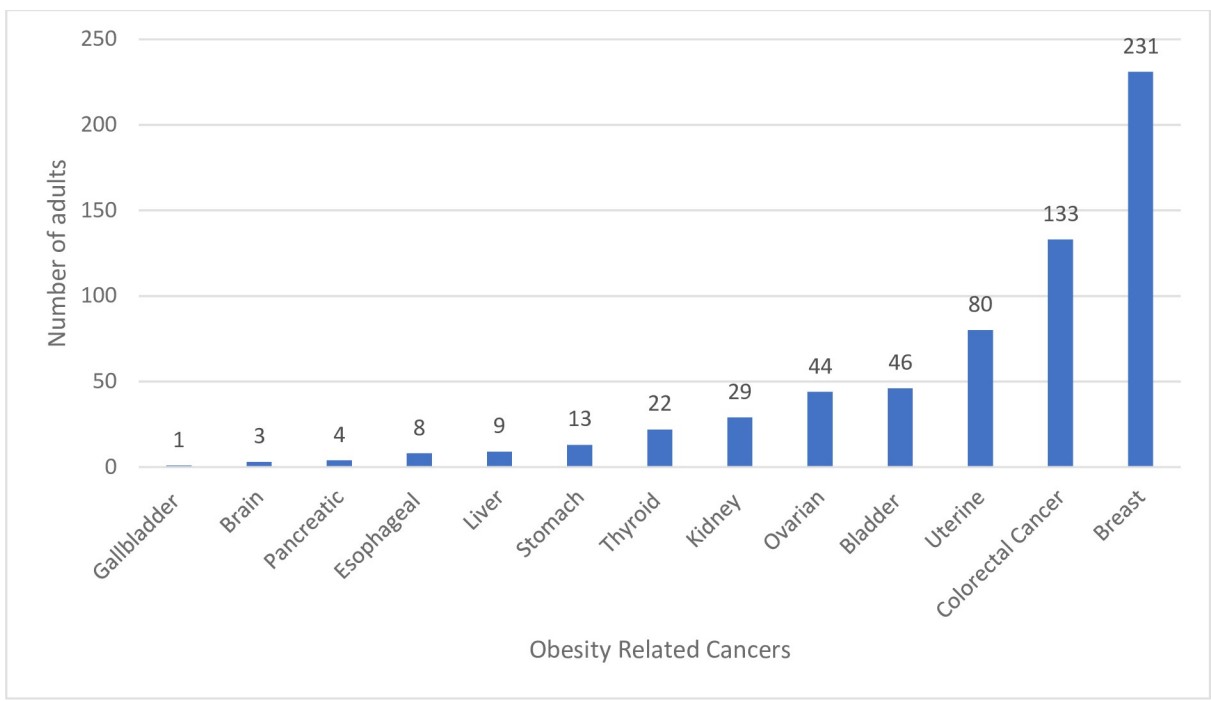

**Fig 1. Prevalence of specific obesity-related cancers.** The most common were breast (14.6%), colorectal (6.6%), and uterine (4.3%) cancer.

<0.0001). For race/ethnicity, non-Hispanic Blacks were 2.1 times more likely to have an ORC than non-Hispanic Whites (95% CI = 1.40–3.38; $p$ = <0.001); and Hispanics were 2.5 times more likely to have an ORC than non-Hispanic Whites (95% CI = 1.39–4.77; $p$ = <0.01). For every 1-year unit increase in age, the odds of ORC increased by 3% (95% CI = 1.00–1.05; $p$ = <0.01). Education, marital status, insurance status, income, and smoking status were not significant predictors of ORC (Table 2).

## Discussion

Evidence consistently shows that higher BMI is associated with an increased risk of ORCs [1,17–22]. From 1997 to 2014, obesity rates rose significantly among adult cancer survivors compared with rates in the general population [23]. In our study, we identified a higher prevalence of ORCs in both non-Hispanic Black and Hispanic study participants (>50% for each) with MetS compared with non-Hispanic Whites with MetS, who had a prevalence of less than one-third. This higher prevalence remained significant after controlling for multiple demographic, socioeconomic, and medical factors.

Our findings are consistent with a subgroup analysis in a previous study examining data from the 2008–2015 Medical Expenditure Panel Survey, which used estimated annual health expenditures by ORCs and other cancer types while controlling for sociodemographic and clinical characteristics.[2] The analysis showed a higher prevalence of ORCs among racial and ethnic minorities (32%–46%) compared with the prevalence among non-Hispanic Whites (25%). Although the overall descriptive analysis identified differences in the prevalence of ORC by race/ethnicity, a regression model was not performed to identify groups at higher risk for developing ORC in individuals with MetS [2]. Similarly, Steele et al. assessed data from several cancer registries by sex, age, race/ethnicity, and U.S. region. Incidence rates for overweight and ORC were higher among non-Hispanic Blacks and non-Hispanic Whites compared with

**Table 1. Baseline characteristics of comorbid metabolic syndrome and cancer, NHANES 1999–2014.**

| Characteristics | All (N = 1554) | Diagnosed with an ORC | | p-value |
| --- | --- | --- | --- | --- |
| | | Yes, % (N = 580) | No, % (974) | |
| Age | | 64.2 (SE ± 0.76) | 67.5 (SE ± 0.46) | <0.001 |
| Sex | | | | <0.001 |
| Female | 788 | 448 (81.6) | 340 (39.9) | |
| Male | 766 | 132 (18.4) | 634 (60.1) | |
| Race | | | | <0.001 |
| Non-Hispanic Black | 239 | 112 (9.84) | 127 (4.6%) | |
| Hispanic | 189 | 101 (7.2) | 88 (3.01)) | |
| Non-Hispanic White | 1126 | 367 (82.9) | 759 (92.38) | |
| Marital status | | | | 0.01 |
| Married | 909 | 286 (57.4) | 623 (67.21) | |
| Divorce | 170 | 73 (12.9) | 97 (11.1) | |
| Living with partner | 37 | 10 (1.2) | 27 (2.8) | |
| Never married | 63 | 28 (4) | 35 (3.8) | |
| Separated | 38 | 12 (1.86) | 26 (1.79) | |
| Widowed | 326 | 166 (45.5) | 160 (54.5) | |
| Education | | | | <0.001 |
| Less than high school | 450 | 210 (25.3) | 240 (16.5) | |
| High school/GED (ref) | 376 | 141 (26.6) | 235 (25.3) | |
| Some college and above | 726 | 229 (48.1) | 497 (58.2) | |
| Annual Income | | | | 0.01 |
| <$35,000 | 404 | 177 (32) | 227 (22.7) | |
| $35,000–74,999 | 264 | 105 (21.7) | 159 (21.9) | |
| $75,000+ | 168 | 41 (14.2) | 127 (22.5) | |
| Missing | 675 | 240 (32.1) | 435 (32.9) | |
| Health Insurance | | | | 0.29 |
| No | 87 | 36 (4.7) | 51 (6.6) | |
| Yes | 1461 | 544 (95.3) | 917 (93.4) | |
| Smoker | | | | 0.5 |
| Every day | 188 | 59 (24.3) | 129 (25) | |
| Some days | 32 | 15 (4.4) | 17 (2.5) | |
| Not at all | 661 | 205 (71.3) | 456 (72.4) | |

Weighted percentages were calculated.

the rates for Hispanics, American Indians/Alaska Natives, and Asians/Pacific Islanders, but again did not conduct a regression model to identify groups at higher risk for developing ORC in individuals with MetS.

Our study identified the increased prevalence of ORC in individuals from the Hispanic and African American communities with MetS. This association warrants increased attention to considering cardiovascular prevention and early intervention strategies in these high risk groups. The explanations for racial/ethnic differences seen in our study and previous studies are not clear. Race and ethnicity are social constructs that mainly reflect inequities due to the longstanding and persisting role of structural racism on health [24]. These inequities could account for the increased cardiovascular risk as well as a lifetime of social and economic stressors that may lead to weathering and be expressed as increased allostatic load, inflammation, epigenetic changes, and more [25,26], that may interact with ORC.

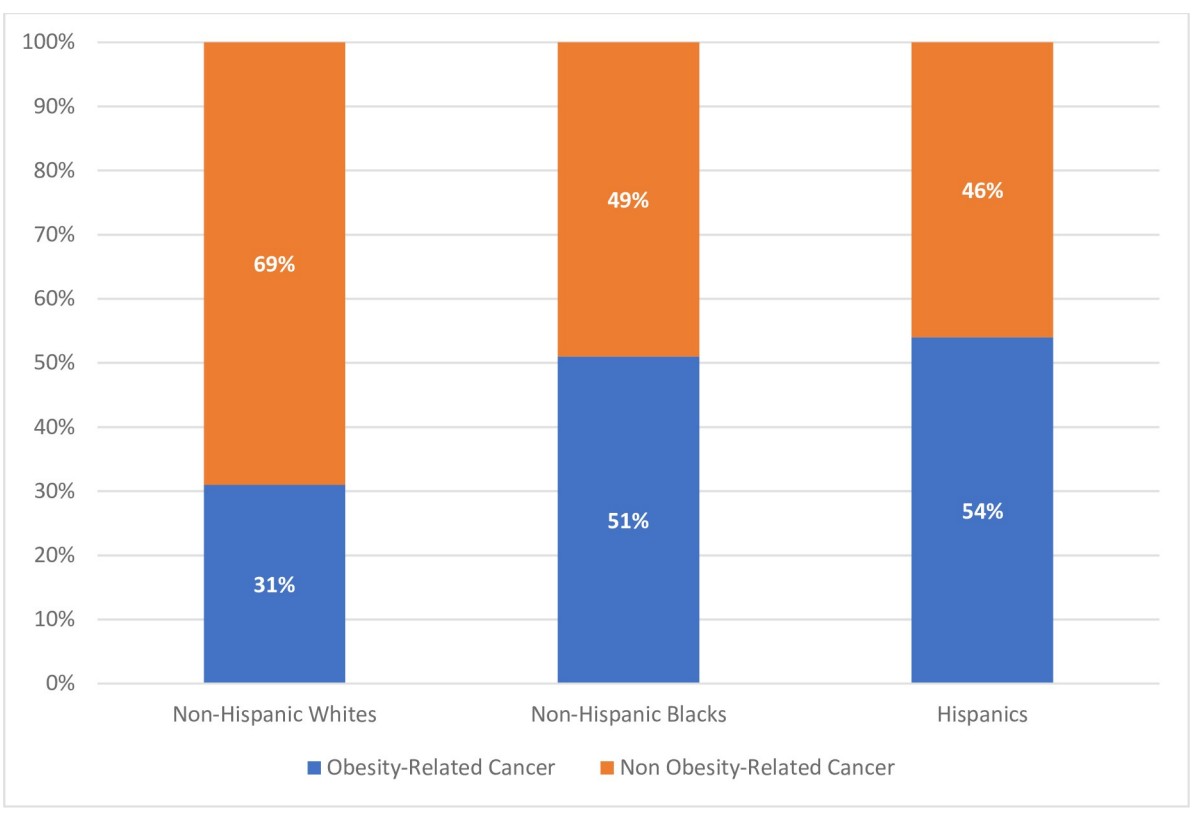

**Fig 2. Prevalence of obesity-related cancers among NHANES participants with metabolic syndrome, by race and ethnicity.** The prevalence of obesity-related cancers was 31% in non-Hispanic Whites, 51% in non-Hispanic Blacks, and 54% in Hispanics ($p$ = <0.001).

Although limited data are available on the direct effects of having an ORC compared to a non-ORC, it has been demonstrated that adults with an ORC had approximately 2.1 times higher excess health care expenditures than those with a non-ORC [2]. Therefore, prevention and early intervention strategies could reduce not only the individual/family burden but the economic burden within the health care system.

Findings from our study also demonstrated gender as a significant factor associated with ORC in our cohort with MetS. Similar to prior studies [2,27], we found that women were more likely than males to be diagnosed with an ORC, potentially due to breast cancer being the most prevalent ORC within this study. The association between BMI and ORC may differ by sex, as Chadid and colleagues [28] reported that ORC risk among women increased with a BMI of at least 26 kg/m$^2$ but at least 28 kg/m$^2$ for men. Therefore, further studies analyzing ORC risk factors by sex while considering BMI are imperative for future studies. We also identified that as age increased, adults with MetS were more likely to have an ORC. Our findings support those of Steele and colleges in demonstrating that incidence rates for co-occurring overweight and ORC were higher among older persons (aged ≥50 years) than among younger persons, and two-thirds of cases occurred among persons aged 50 to 74 years [1]. By contrast, Hong et al. [2] reported patients with ORC were more likely to be younger, but their analysis was conducted on cancer registry data unlike the study by Steele et al. and our study which analyzed the general population samples.

This study had several limitations. The first limitation was the use of self-reported data on ORC as well as race and ethnicity which could lead to potential bias or inaccurate information. A second limitation was that our study explored all cancers, so we were unable to identify

**Table 2. Multivariable logistic regression analysis of obesity-related cancers in adults with metabolic syndrome, National Health and Nutrition Examination Survey 1999–2014.**

| Independent Variable | OR (95% CI) | p-value |
|---|---|---|
| Age | 1.03 (1.0–1.05) | <0.01 |
| Sex | | |
| Female | 6.27 (3.89–10.12) | <0.0001 |
| Male (ref.) | - | - |
| Race | | |
| Non-Hispanic Black | 2.1 (1.40–3.38) | <0.001 |
| Hispanic | 2.5 (1.39–4.77) | <0.01 |
| Non-Hispanic White (ref.) | - | - |
| Marital status | | |
| Married | 1.81 (0.96–3.41) | 0.06 |
| Divorce | 2.07 (0.92–4.67) | 0.07 |
| Living with partner | 0.58 (0.14–2.35) | 0.44 |
| Never married | 1.72 (0.45–6.51) | 0.41 |
| Separated | 0.69 (0.20–2.31) | 0.54 |
| Widowed (ref.) | - | |
| Education | | |
| Less than high school | 1.63 (0.97–2.73) | 0.07 |
| High school/GED (ref) | - | |
| Some college and above | 1.28 (0.68–2.40) | 0.98 |
| Annual Income | | |
| <$35,000 | 1.15 (0.53–2.48) | 0.70 |
| $35,000–74,999 | 0.66 (0.29–1.56) | 0.35 |
| Missing | 0.66 (0.31–1.42) | 0.29 |
| $75,000+ (ref) | - | - |
| Health Insurance | | |
| No | 0.44 (0.13–1.54) | 0.20 |
| Yes (ref) | - | - |
| Smoker | | |
| Everyday | 0.84 (0.25–2.83) | 0.81 |
| Not at all | 0.84 (0.26–2.73) | 0.81 |
| Some days (ref) | - | - |

OR = odds ratio. The regression controlled for race, sex, income, insurance, education, marital status, and smoking status.

whether a specific cancer dominated our findings. Our approach provided us with general information. Future studies can address more specific information. A third limitation was our inability to control for additional factors such as access to care; however, over 90% of our cohort had health insurance. A fourth limitation was that we could not capture all ORCs, such as multiple myeloma and meningioma, due to the broad category of blood cancers or brain cancer. However, numbers for brain and blood cancers were extremely low within the dataset; therefore, we could predict this would not impact our current findings. These limitations were balanced by the many strengths of our study, including a nationally represented cohort undergoing a rigorous health analysis in a structured manner and the ability to control for demographic, socioeconomic, and clinical data. Thus, our finding strengthens the current literature on race/ethnicity differences in ORC prevalence, especially in a high-risk cohort with MetS.

Future directions and interventions could focus on the impact of MetS treatment (e.g., weight loss education, diet, and exercise) on ORC, as well as to examine if these associations are driven by individual components of MetS. This would provide us with additional steps to understanding CVD mortality differences by race/ethnicity within the ORC population. In summary, obesity is the second most common prevention of cancer, and in study participants with MetS we found the adjusted prevalence of ORC to be higher in non-Hispanic Blacks and Hispanics than among non-Hispanic Whites. Therefore, strategies to increase awareness for cancer risk among non-Hispanic Blacks and Hispanics with MetS are warranted, as are prospective studies to determine whether MetS treatment can reduce ORC risk.

## Acknowledgments

We would like to thank Gina Hamilton for project administration and Sharese Terrell Willis, PhD, for editing assistance.

## Author Contributions

**Conceptualization:** Shanada Monestime, Bettina Beech, Dulcie Kermah, Keith Norris.

**Formal analysis:** Dulcie Kermah.

**Funding acquisition:** Shanada Monestime, Bettina Beech, Keith Norris.

**Methodology:** Shanada Monestime.

**Software:** Dulcie Kermah.

**Supervision:** Bettina Beech, Keith Norris.

**Validation:** Shanada Monestime.

**Writing – original draft:** Shanada Monestime.

**Writing – review & editing:** Shanada Monestime, Bettina Beech, Dulcie Kermah, Keith Norris.

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
