## [Decision Letter · Decision Letter 0]

8 Feb 2021

PONE-D-20-37144

Prevalence and Predictors of Obesity-Related Cancers Among Racial/Ethnic Groups with Metabolic Syndrome

PLOS ONE

Dear Dr. Monestime,

Thank you for submitting your manuscript to PLOS ONE. After careful consideration, we feel that it has merit but does not fully meet PLOS ONE’s publication criteria as it currently stands. Therefore, we invite you to submit a revised version of the manuscript that addresses the points raised during the review process.

An expert in the field handled your manuscript, and we are appreciative of their time and contributions. Although some interest was found in your study, several major comments arose that require your attention. Notably, there are questions about the novelty of this study and the presentation of the statistical analyses; more specifics about the patients need to be included; and the discussion section should be utilized to place the current findings into perspective of the existing literature. Please address ALL of the reviewer's comments in your revised manuscript and outline all changes in a response-to-reviewers document.

We look forward to receiving your revised manuscript.

Kind regards,

Frank T. Spradley

Academic Editor

PLOS ONE

Reviewers' comments:

Reviewer's Responses to Questions

**Comments to the Author**

1. Is the manuscript technically sound, and do the data support the conclusions?

Reviewer #1: Partly

2. Has the statistical analysis been performed appropriately and rigorously? 

Reviewer #1: Yes

3. Have the authors made all data underlying the findings in their manuscript fully available?

Reviewer #1: Yes

4. Is the manuscript presented in an intelligible fashion and written in standard English?

Reviewer #1: Yes

5. Review Comments to the Author

Reviewer #1: In manuscript entitled ‘Prevalence and Predictors of Obesity-Related Cancers Among Racial/Ethnic Groups with Metabolic Syndrome’ by Shanada Monestime et al., authors observe that obesity related cancers are more prevalent in Blacks and Hispanics, males and younger adults with metabolic syndrome.

I have several reservations for the study. My specific comments are appended as below:

1. Race and ethnicity is well known to be associated with the cancer aggressiveness. For instance, the death rate in African American females is 14% higher despite 7% low incidence rate (PMID: 30558195). There seems less novelty in the study reported.

2. While referring to the cardiovascular disease, authors cite relatively old reference while there are many recent evidences (PMID: 31761945, PMID: 28624099, PMID: 28421481). A careful review of cited literature is essential.

3. The authors claims that the data was obtained from 1999-2014 while the reference (reference no 13) they cite is published in 1994. This need to be explained.

4. Does authors specify past vs current individuals under medication for diabetes/hypertension?

5. While accessing the smoking status, does authors pay attention towards past/current smoker? It may not be relevant to classify a past smoker who at present does not smoke to classify in the third criteria (not at all a smoker).

6. Statistical inference should be quoted in the figures (Figure 1, 2).

7. As authors notes the observations in a range of cancer, it should be specified which cancer the obesity and metabolic syndrome related observations meet the statistical significance and what message authors like to spread through it?

8. Which cancers authors find the gender related disparity? How does authors reconcile it with hormone status?

9. As authors use only one source for the data, once the possible role of factors is identified, a careful review of literature could have conducted to justify the observations. For instance, the gender/ obesity related disparity among the observed cancers.

6. PLOS authors have the option to publish the peer review history of their article (what does this mean?). If published, this will include your full peer review and any attached files.

Reviewer #1: **Yes: **Ravindra Deshpande

---

## [Author Response · Author response to Decision Letter 0]

11 Mar 2021

1. Reviewer’s comment: Race and ethnicity is well known to be associated with the cancer aggressiveness. For instance, the death rate in African American females is 14% higher despite 7% low incidence rate (PMID: 30558195). There seems less novelty in the study reported.

a. Response: Although it is known for African Americans to have high rates of several types of cancers, our study provides novelty because we performed a regression model to further understand if after considering sociodemographic factors, did racial differences exist with obesity related cancers? Interestingly, when we divided the cancers by obesity versus non-obese cancers, African Americans were linked to having higher rates of Obesity Related Cancers and Whites were linked to having higher rates of Non-Obesity Related Cancers. 

2. Reviewer’s comment While referring to the cardiovascular disease, authors cite relatively old reference while there are many recent evidences (PMID: 31761945, PMID: 28624099, PMID: 28421481). A careful review of cited literature is essential.

a. Response The reviewer brought up a great point. Within my literature search, I have come across several recent articles between 2017-2020. The issue I am encountering is when I want to reference something from the newer articles, the statement within those articles refers back to another article (the original source) which are backdated to 2010-2014. However, I was able to update one of my references to reflect a more updated finding. Examples are below:

b. Response Although reference 9 dates back from 2010, this was a review article that discussed the pathophysiology of Mets as it relates to cancer. Several new studies refers back to this article so I used it to support my statements from primary literature. 

3. Reviewer’s comment The authors claims that the data was obtained from 1999-2014 while the reference (reference no 13) they cite is published in 1994. This need to be explained.

a. Response Thank you for identifying this. I updated the references to reflect 1999-2014

4. Reviewer’s comment Does authors specify past vs current individuals under medication for diabetes/hypertension?

a. Response Within the study population section, we included key words to indicate that survey respondents had to currently be on treatment in order for us to identify if they had hypertension and/or diabetes. For example the questions regarding medications were: 

i. Are you now taking prescribed medicine (for blood pressure)?”

ii. “Are you now taking diabetic pills to lower blood sugar?” 

iii. Are you taking insulin now?”

b. Response We did not include patients with a past medication history because we wanted to identify current status of diabetes and/or hypertension. In this section we also assessed lab values to determine diabetes and/or hypertension. 

5. Reviewer’s comment While accessing the smoking status, does authors pay attention towards past/current smoker? It may not be relevant to classify a past smoker who at present does not smoke to classify in the third criteria (not at all a smoker).

a. Response Thank you for your comment. We classified smoking status in three categories: smoking every day, some days, or not at all. Past smoking was categorized in the (not at all a smoker). Current smokers were broken down into the following two categories: smoking every day, some days. Do you mind providing clarity on the request and I am happy to update this portion of the manuscript?

6. Reviewer’s comment Statistical inference should be quoted in the figures (Figure 1, 2).

a. Response The statistical inference for Figures 1 and 2 are now located under the image. For figure 1, our statistical analysis focused on the frequencies.

7. Reviewer’s comment As authors notes the observations in a range of cancer, it should be specified which cancer the obesity and metabolic syndrome related observations meet the statistical significance and what message authors like to spread through it?

a. Response Thank you for this great comment. Our current funding supported a statistician to support one specific aim. Our aim was to assess if racial differences exist regarding the risks of having an obesity related cancer for patients with metabolic syndrome. Our study revealed that there is a significant risk difference by race. Therefore, we are submitting a second small grant to get funding for a statistician to assist us with determining if differences of ORC exist by gender and cancer type, while stratifying by race. Although we are unable to address cancer type into our current regression model, this manuscript serves as a foundation to demonstrate and support the next steps to identify differences by cancer types. 

b. Response I removed comments regarding breast and colon cancer from the discussion due to this being an observation/speculation and not supported by our statistical analysis. This redirects our manuscript to focus on our main findings. Thank you again for the great feedback. 

8. Reviewer’s comment Which cancers authors find the gender related disparity? How does authors reconcile it with hormone status?

a. Response Thank you for your excellent comment. Similar to the statement above, during our research, our primary aim focused on if racial differences exist in the risk of developing an ORC in the Mets population. Our study demonstrated that gender differences exist. Therefore, within our next small grant application we now have a foundation to support further assessing gender differences as it related to different cancer types

9. Reviewer’s comment As authors use only one source for the data, once the possible role of factors is identified, a careful review of literature could have conducted to justify the observations. For instance, the gender/ obesity related disparity among the observed cancers.

a. Response Thank you for your comment. We revised the discussion to focus on our primary aim which was to identify if racial differences exist in the risk level of developing an ORC while confounding for several factors which prior studies did not do. Research within the Mets and ORC space by race/ethnicity is limited, but we were able to identify two main studies Steele et al and Hong et al which are listed within our discussion. that supports our findings. Within these studies they did not conduct a regression model to consider confounders so it was difficult to truly know if race/ethnicity played a role. However, how study strengthen the current findings because we performed a regression model with pertinent confounders which confirms, even with confounders in the model, racial/ethnicity differences still exist. 

b. Response In regards to gender differences, we were unable to find anything in Steele or Hong et al. however, we mentioned how we could use information from Chadid et al. to further analyze this data by BMI, which we plan on conducting for our second study. We needed justification as to why studying gender differences by cancer type is critical and believe our findings will serve as a foundation.

---

## [Decision Letter · Decision Letter 1]

15 Mar 2021

Prevalence and Predictors of Obesity-Related Cancers Among Racial/Ethnic Groups with Metabolic Syndrome

PONE-D-20-37144R1

Dear Dr. Monestime,

We’re pleased to inform you that your manuscript has been judged scientifically suitable for publication and will be formally accepted for publication once it meets all outstanding technical requirements.

Kind regards,

Frank T. Spradley

Academic Editor

PLOS ONE

Reviewers' comments:

Reviewer's Responses to Questions

**Comments to the Author**

1. If the authors have adequately addressed your comments raised in a previous round of review and you feel that this manuscript is now acceptable for publication, you may indicate that here to bypass the “Comments to the Author” section, enter your conflict of interest statement in the “Confidential to Editor” section, and submit your "Accept" recommendation.

Reviewer #1: All comments have been addressed

2. Is the manuscript technically sound, and do the data support the conclusions?

Reviewer #1: Partly

3. Has the statistical analysis been performed appropriately and rigorously? 

Reviewer #1: Yes

4. Have the authors made all data underlying the findings in their manuscript fully available?

Reviewer #1: Yes

5. Is the manuscript presented in an intelligible fashion and written in standard English?

Reviewer #1: Yes

6. Review Comments to the Author

Reviewer #1: Authors have now answered all my comments and manuscript in present form is much better. I approve it for publication.

7. PLOS authors have the option to publish the peer review history of their article (what does this mean?). If published, this will include your full peer review and any attached files.

Reviewer #1: **Yes: **Ravindra Pramod Deshpande

---

## [Editor Report · Acceptance letter]

22 Mar 2021

PONE-D-20-37144R1 

Prevalence and Predictors of Obesity-Related Cancers Among Racial/Ethnic Groups with Metabolic Syndrome 

Dear Dr. Monestime:

I'm pleased to inform you that your manuscript has been deemed suitable for publication in PLOS ONE. Congratulations! Your manuscript is now with our production department. 

Kind regards, 

on behalf of

Dr. Frank T. Spradley 

Academic Editor

PLOS ONE